# Ibuprofen, Flurbiprofen, Etoricoxib or Paracetamol Do Not Influence ACE2 Expression and Activity In Vitro or in Mice and Do Not Exacerbate In-Vitro SARS-CoV-2 Infection

**DOI:** 10.3390/ijms23031049

**Published:** 2022-01-19

**Authors:** Natasja de Bruin, Ann-Kathrin Schneider, Philipp Reus, Sonja Talmon, Sandra Ciesek, Denisa Bojkova, Jindrich Cinatl, Imran Lodhi, Bruce Charlesworth, Simon Sinclair, Graham Pennick, William F. Laughey, Philip Gribbon, Aimo Kannt, Susanne Schiffmann

**Affiliations:** 1Fraunhofer Institute for Translational Medicine and Pharmacology ITMP, Theodor-Stern-Kai 7, 60596 Frankfurt am Main, Germany; Natasja.Debruin@itmp.fraunhofer.de (N.d.B.); Ann-Kathrin.Schneider@itmp.fraunhofer.de (A.-K.S.); sonja.talmon@itmp.fraunhofer.de (S.T.); Sandra.Ciesek@kgu.de (S.C.); 2Fraunhofer Cluster of Excellence Immune Mediated Diseases, CIMD, 60596 Frankfurt am Main, Germany; 3Institute of Medical Virology, University Hospital Frankfurt/Main, Goethe University, Paul-Ehrlich-Str. 40, 60596 Frankfurt am Main, Germany; Philipp.Reus@itmp.fraunhofer.de (P.R.); Denisa.Bojkova@kgu.de (D.B.); cinatl@em.uni-frankfurt.de (J.C.); 4Fraunhofer Institute for Translational Medicine and Pharmacology ITMP, Discovery Research ScreeningPort, 22525 Hamburg, Germany; Philip.Gribbon@itmp.fraunhofer.de; 5Reckitt Healthcare Ltd., Dansom Lane South, Kingston Upon Hull HU8 7DS, UK; Imran.Lodhi@rb.com (I.L.); Bruce.Charlesworth@rb.com (B.C.); Simon.Sinclair@rb.com (S.S.); Graham.Pennick@rb.com (G.P.); Bill.Laughey@rb.com (W.F.L.); 6Health Professions Education Unit, Hull York Medical School, University of York, Heslington, York YO10 5DD, UK; 7Pharmazentrum Frankfurt/ZAFES, Department of Clinical Pharmacology, Goethe-University Hospital Frankfurt am Main, Theodor-Stern-Kai 7, 60590 Frankfurt am Main, Germany

**Keywords:** NSAIDs, ibuprofen, ACE2 mRNA expression, ACE protein expression, ACE2 activity, SARS-CoV-2 infection

## Abstract

SARS-CoV-2 uses the human cell surface protein angiotensin converting enzyme 2 (ACE2) as the receptor by which it gains access into lung and other tissue. Early in the pandemic, there was speculation that a number of commonly used medications—including ibuprofen and other non-steroidal anti-inflammatory drugs (NSAIDs)—have the potential to upregulate ACE2, thereby possibly facilitating viral entry and increasing the severity of COVID-19. We investigated the influence of the NSAIDS with a range of cyclooxygenase (COX)1 and COX2 selectivity (ibuprofen, flurbiprofen, etoricoxib) and paracetamol on the level of ACE2 mRNA/protein expression and activity as well as their influence on SARS-CoV-2 infection levels in a Caco-2 cell model. We also analysed the ACE2 mRNA/protein levels and activity in lung, heart and aorta in ibuprofen treated mice. The drugs had no effect on ACE2 mRNA/protein expression and activity in the Caco-2 cell model. There was no up-regulation of ACE2 mRNA/protein expression and activity in lung, heart and aorta tissue in ibuprofen-treated mice in comparison to untreated mice. Viral load was significantly reduced by both flurbiprofen and ibuprofen at high concentrations. Ibuprofen, flurbiprofen, etoricoxib and paracetamol demonstrated no effects on ACE2 expression or activity in vitro or in vivo. Higher concentrations of ibuprofen and flurbiprofen reduced SARS-CoV-2 replication in vitro.

## 1. Introduction

SARS-CoV-2 uses angiotensin converting enzyme 2 (ACE2) for cell entry after spike protein priming by the transmembrane protease serine 2 (TMPRSS2) [1]. Patients with COVID-19 are more likely to have a serious infection, or to die, if they also have comorbidities, including cardiovascular and cerebrovascular disease, conditions commonly treated with ACE inhibitors (ACEi) or angiotensin receptor blockers (ARBs) [2,3,4]. Based on a reported assumption that ACEi or ARBS, but also non-steroidal anti-inflammatory drugs (NSAIDs) like ibuprofen, may increase ACE2 expression [4], it was hypothesized that these drugs may promote SARS-CoV-2 cell entry and exacerbate COVID-19 [5]. This has sparked a controversy, especially about the use of ibuprofen as an analgesic and antipyretic therapy in COVID-19 [6,7].

ACE converts angiotensin I into angiotensin II, which mediates vasoconstriction and blood pressure-increasing effects via the angiotensin 1 (AT_1_)-receptor and the Renin Angiotensin Aldosterone System (RAAS). Consequently, ACE inhibitors and AT_1_-receptor antagonists are frequently used for patients suffering from hypertension. ACE2, in turn, converts angiotensin II into angiotensin 1–7 and intervenes in a counter-regulatory manner in the cardiovascular system and RAAS by reducing the angiotensin II concentration. ACE2 is a membrane bound aminopeptidase which can be cleaved from the membrane, thereby generating a soluble form of ACE2 [8]. Interestingly, ACE2 is highly expressed on the surface of heart and lung cells, the latter being used by coronaviruses such as SARS-CoV and SARS-CoV-2 to enter host cells [1]. The theoretical relationship between SARS-CoV-2 infection, ACE2 levels and the RAAS is complex and incompletely understood: on the one hand it is hypothesized that up-regulating ACE2 can facilitate viral entry [1], but on the other hand it is proposed that SARS-CoV-2 itself downregulates ACE2 leading to unopposed RAAS activation which is associated with pro-inflammatory damage to lung and other tissue [9]. If this second hypothesis is correct, drugs such as ACE-inhibitors, which inhibit the RAAS, could be protective in COVID-19 [10,11]. Furthermore, it has been theorized that because COVID-19 is a pro-inflammatory disease, complicated in its severe form by cytokine storms, then modulation of the immune system by NSAIDs, which reduce inflammatory mediators by inhibiting cyclooxygenase (COX), may have a protective effect [11,12]. This hypothesis is being tested with the LIBERATE trial (https://clinicaltrials.gov/ct2/show/study/NCT04334629; 17 December 2021), in which hospitalized patients in respiratory failure are receiving ibuprofen or placebo in addition to standard care.

Given the theoretical concerns about the upregulation of ACE2 expressed early in the pandemic [5] and given that evidence for an association between ibuprofen and the regulation of ACE2 expression is scarce, we designed a study to address this research gap. The aim of this study is to investigate whether ibuprofen influences ACE2 protein/mRNA expression or activity in vitro and in mice. We also aim to investigate whether ibuprofen affects SARS-CoV-2 replication in vitro. Furthermore, to evaluate any class effect for NSAIDs, flurbiprofen and etoricoxib were also studied. Placebo and a relevant non-NSAID analgesic, paracetamol, were used as comparators. The choice of NSAIDs was based on obtaining a broad range of COX1 and COX2 inhibition: etoricoxib is highly COX2 selective, while ibuprofen and flurbiprofen inhibit both COX1 and COX2, with flurbiprofen demonstrating the highest COX1 inhibition of the three NSAIDs [13]. The various concentrations of drugs used in the in vitro studies, and the various doses used in mice, were selected with the intention of spanning the range of pharmacologically relevant therapeutic levels in humans. For the in vitro studies, Caco-2 cells were chosen on the basis that they are highly permissive for SARS-CoV-2 virus infection [10] and express both ACE2 and COX2.

## 2. Results

### 2.1. ACE2 mRNA and Protein Expression Is Not Regulated by NSAIDs and Paracetamol

First, we investigated whether ibuprofen regulates ACE2 mRNA and/or protein expression. For that purpose, Caco-2 cells were incubated with increasing drug concentrations for 24 h and 48 h, with the selected concentrations being in the range of c_max_ plasma levels in humans [14]. Ibuprofen did not alter ACE2 mRNA and protein expression (Figure 1A,B). We additionally tested flurbiprofen, paracetamol and etoricoxib in relevant concentrations [15,16,17]. Paracetamol and etoricoxib significantly reduced the ACE2 mRNA expression, but had no effect on ACE2 protein expression. Flurbiprofen did not alter ACE2 mRNA expression but slightly increased ACE2 protein expression (Figure 1A,B). Due to the small effects, which are presumably not biologically relevant, these data indicate that the investigated analgesics do not modulate the efficiency of SARS-CoV-2 entry into Caco-2 cells by means of regulation in ACE2 levels.

### 2.2. ACE2 Activity Is Not Regulated by NSAIDs and Paracetamol

We investigated whether the test compounds influence the activity of soluble and/or membrane-bound ACE2 in Caco-2 cells. The ACE2 inhibitor MLN-4760 was used as positive control and inhibited soluble and membrane-bound ACE2 with an IC_50_ value of 4 and 26.8 nM, respectively. However, ibuprofen, flurbiprofen, etoricoxib and paracetamol did not influence the activity of the soluble and membrane-bound ACE2 (Figure 2).

### 2.3. Ibuprofen and Flurbiprofen Reduced Infection Potential of SARS-CoV-2 at High Concentrations

To investigate whether the tested drugs might promote or inhibit SARS-CoV-2 infection, Caco-2 cells were pre-incubated for 1 h with the test drugs and subsequently infected with SARS-CoV-2 at a multiplicity of infection (MOI) of 0.01 for 24 h. The infection level of Caco-2 cells by SARS-CoV-2 was determined via immunochemical staining of the SARS-CoV-2 spike protein. The dose ranges tested for the different drugs were the same as in the ACE2 expression and activity tests, and were selected to include the range of c_max_ plasma levels in humans (*vide supra*). Values of 438.1 and 219 µM ibuprofen as well as 204.7 µM flurbiprofen significantly reduced viral load in Caco-2 cells, while etoricoxib and paracetamol had no consistent, significant effect on viral load (Figure 3).

### 2.4. ACE2 mRNA and Protein Expression in Murine Lung, Heart and Aorta Is Not Influenced by Ibuprofen

To confirm our in vitro results, we tested whether ibuprofen affects ACE2 mRNA/protein expression in mice. We investigated ACE2 expression in lung, heart and aorta to address the SARS-CoV-2 and drug relevant tissues. C57BL6 mice were treated with different doses of ibuprofen (0, 50, 100, 200 mg/kg) for 7 days (for details see Appendix A). Ibuprofen dose selection was based on previous studies in mice and clinically recommended doses [18,19]. We observed no differences in relative ACE2 mRNA levels in lung, heart and aorta tissue with ibuprofen treatment (Figure 4A, Appendix A). There was also no difference in the relative ACE2 protein levels in the addressed tissues in ibuprofen treated mice in comparison to untreated mice (Figure 4B, Appendix A). These data indicate that ibuprofen did not influence ACE2 protein and mRNA expression in mice.

### 2.5. ACE2 Activity in Plasma Is Not Altered by Ibuprofen Treatment

To investigate whether ibuprofen influences ACE2 activity in vivo, the product of ACE2 (angiotensin 1–7) and the substrate angiotensin II plasma concentrations in ibuprofen treated mice were analysed. The concentration of angiotensin 1–7 in untreated mice was about 50 pg/ml. The treatment with ibuprofen did not significantly alter the angiotensin 1–7 concentrations (Figure 4C, Appendix A). The concentration of angiotensin II was below the detection limit (<0.1 pg/mL) of the used ELISA in both untreated and treated mice. These results confirm our in vitro data that ibuprofen does not alter ACE2 activity.

## 3. Discussion

We demonstrated that ACE2 mRNA/protein expression and activity is not modulated by the here tested NSAIDs ibuprofen, flurbiprofen, etoricoxib or paracetamol in Caco-2 cells. Our results revealed that these drugs have no effect on the ACE2 mRNA and protein expression in the colon epithelial cell line Caco-2. This is in line with data indicating that about 44 µM ibuprofen did not induce ACE2 mRNA expression in the lung cancer cell line Calu-3 and in the hepatic cancer cell line Huh7.5 [12].

Our in vivo data also showed no increase of ACE2 mRNA and protein expression in ibuprofen treated healthy mice, which is in accordance with Chen at al., who also found no evidence that either ibuprofen or meloxicam upregulate ACE2 expression in mice—wildtype mice treated for three days with ibuprofen (30 mg/kg, i.p.) or meloxicam (1 mg/kg) showed no altered ACE2 mRNA expression in lung, heart, kidney and ileum [12]. In contrast to this, data from other research groups indicates that ibuprofen increases ACE2 expression in diseased and healthy rats. A total of 40 mg/kg ibuprofen administered to streptozotocin-induced diabetic rats over a period of 8 weeks led to an increase of ACE2 mRNA and protein expression in myocardial tissue [20]. A total of 40 mg/kg ibuprofen (administered for 3 weeks) increased lung ACE2 mRNA and protein expression in metabolic syndrome rats [21]. However, diabetic and metabolic syndrome rats were characterized by a reduced ACE2 expression in comparison to normal rats [20,21] indicating that ibuprofen prevented the reduction of ACE2 expression in diseased rats. In healthy rats, treatment with 40 mg/kg ibuprofen over three weeks increased lung ACE2 protein expression and activity [21]. However, in the same study, ibuprofen treatment led to a decrease in spike protein internalization in A549 pneumocytes. The varying results of the in vivo models may be explained by the difference of the exposure times to the compound and the different rodents. The mice in our study were treated for 1 week with ibuprofen, whereas the healthy, diabetic and metabolic syndrome rats were treated for 3 and 8 weeks, respectively [20,21].

Our data indicate that both ibuprofen and flurbiprofen reduce SARS-CoV-2 load in Caco-2 cells, but only at higher tissue concentrations (219 µM and 438 µM for ibuprofen and 205 µM for flurbiprofen) (Figure 3). Of course, it is difficult to know how in vitro findings in Caco-2 cells would translate into clinical effects, but concentrations of these drugs in body tissues at therapeutic doses clearly merit consideration. Although much is known about plasma levels, little is known about levels in non-plasma tissue. NSAIDs are highly protein bound, so we might expect plasma levels to be generally higher than those in other tissues. In a study of patients undergoing knee arthroplasty due to osteoarthritis, Gallelli et al. measured concentrations of ibuprofen in plasma and in synovial fluid after 7 days of dosing—for patients taking 1800 mg daily, the plasma concentration was approximately 200 µg/mL (≈876 µM) and the synovial fluid concentration was approximately 50 µg/mL (≈219 µM) [22]. In the case of flurbiprofen, Kai et al. administered two relatively low oral doses of 40 mg per dose at 16 h and 2 h before cruciate surgery and measured concentrations in plasma and various soft tissues—levels were 3.3 µg/mL (≈14.5 µM) in plasma and 0.08 µg/g in muscle (≈0.4 µM) [23]. COVID-19 is a multisystem disease affecting many tissues, though the target organ for initial viral entry and replication is lung. Given that lung tissue is not a therapeutic target for NSAIDs in normal use, no data are available for lung concentrations. It is therefore very difficult to know if the concentrations required to reduce viral load in vitro in this study can be achieved in relevant body tissues in patients at therapeutic doses. For flurbiprofen, this seems unlikely. For ibuprofen, if synovial fluid levels are a guide, which is speculative, then it may be possible at higher therapeutic doses.

We are aware of one previous study that has exposed cell lines to ibuprofen and assessed viral load (Chen et al. [12]). In this study, the cells were primary human bronchial epithelial cells (HBECs) and ibuprofen was tested at a relatively low concentration (50 µM) and had no effect on viral load. This is in line with our data that show viral load only decreases with high concentrations of ibuprofen.

As it occurs at concentrations well above the IC50 for COX, the effect of ibuprofen on viral load is likely due to off target effects of the drug. For example, PARP cleavage has been observed in colon cancer cells (HCT-15 and HCA-7) at high concentrations of ibuprofen [24]. The effect seems to be at least in part COX2 independent, because it was observed also in COX2 negative HCT-15 cells. TMPRSS2 has a poly(ADP-ribose) polymerase 1 (PARP1) binding site near to the promoter region. Since PARP1 regulates gene expression at the transcription level, Lodhi et al. speculated that virus entry can be blocked by inhibiting PARP1 [25]. It is possible that PARP1 reactivates the transcription of TMPRSS2 during viral infection [26]. Thus, ibuprofen may prevent virus entry via inhibition of PARP. This speculation is supported by the finding that the PARP inhibitor stenoparib inhibits SARS-CoV-2 replication in vitro [27].

Several SARS-CoV-2 treatment strategies were described such as blocking of virus entry, targeting cellular signalling pathways, targeting viral RNA-dependent RNA polymerase and interfering with the endocytic pathway. Some of these pathways were identified as targets of repurposed drugs such as telmisartan, sirolimus, remdesivir and chloroquine (for more detail see reviews from Nitulescu et al. [28] and Dehelean et al. [29]). One of the targets for blocking cell entry is the nucleoprotein of SARS-CoV-2. Interestingly, naproxen, which shows structural similarity to ibuprofen, has weak anti-viral effects that were attributed to binding to the nucleoprotein of SARS-CoV-2 [30] and it could be speculated a similar mechanism also applies for ibuprofen.

Ultimately, clinical studies are necessary to gain insights into the effects of NSAIDs on COVID-19. In a prospective cohort study of over 70,000 patients, Drake et al. demonstrated that exposure to ibuprofen or other NSAIDs within 2 weeks of admission for COVID-19 had no effect on the requirement for oxygen, the need for invasive ventilation, acute kidney injury, admission to intensive care or mortality [31]. A recently published meta-analysis of 19 original studies, providing information on exposure to NSAIDs and COVID-19 outcomes, identified that there was no excess risk of acquiring COVID-19 in patients exposed to NSAIDs [32]. Furthermore, for patients infected with SARS-CoV-2, NSAID exposure did not increase the risk of hospital admissions, severe outcomes or death [32]. The review concluded that “the theoretical risks of NSAIDs or ibuprofen in SARS-CoV-2 infection are not confirmed by observational data” [32].

In conclusion, in line with previous research [12], we observed no influence of the here tested NSAIDs (ibuprofen, flurbiprofen, etoricoxib) and paracetamol on ACE2 expression/activity in vitro or in vivo, further refuting previous concerns that NSAIDs may upregulate ACE2 and facilitate SARS-CoV-2 entry [5]. However, we cannot exclude the possibility that other NSAIDs may affect ACE2 expression and activity. At lower concentrations, ibuprofen, flurbiprofen, etoricoxib and paracetamol had no effect on viral replication in vitro. At higher concentrations, ibuprofen and flurbiprofen reduced SARS-CoV-2 load in vitro, whereas paracetamol and the COX2 selective NSAID etoricoxib had no significant effect on viral load.

The strength of our study is that we compared the effect of ibuprofen on ACE2 expression/activity in vitro and in mice, which allows us to conclude that ibuprofen does not facilitate virus entry via an upregulation of ACE2. A further strength is that we compared across a broad range of drug concentrations which allowed us to detect the weak in vitro antiviral activity at higher concentrations. However, the limitation of our study is the lack of clinical data. Therefore, whether this has any significance for patients with COVID-19 can only be established through clinical trials.

## 4. Materials and Methods

### 4.1. Cells and Reagents

Caco-2 cells were purchased from Sigma Aldrich (Schnelldorf, Germany) and cultured in Minimum Essential Medium Eagle medium supplemented with 10% fetal calf serum (FCS) (Thermo Fisher Scientific, Schwerte, Germany), 2 mM L-glutamine (Thermo Fisher Scientific, Schwerte, Germany) and 1× non-essential amino acid solution (Sigma Aldrich, Schnelldorf, Germany). All media contained 1% penicillin/streptomycin (Thermo Fisher Scientific, Schwerte, Germany), and cells were cultured at 37 °C in a 5% CO_2_ atmosphere. Ibuprofen was obtained from Sigma Aldrich (Schnelldorf, Germany) and flurbiprofen, etoricoxib and paracetamol were purchased from Biomol (Hamburg, Germany). Ibuprofen was dissolved in water. For in vitro experiments, flurbiprofen, etoricoxib and paracetamol were dissolved in 100% dimethyl sulfoxide (DMSO) (Sigma Aldrich, Schnelldorf, Germany) and further diluted in media (c_stock_ = 25 mg/mL (for flurbiprofen and paracetamol), c_stock_ = 20 mg/mL for etoricoxib; maximum DMSO concentration during experiments 0.1% (etoricoxib, paracetamol); 0.2% (flurbiprofen)). To exclude DMSO effects in the infection experiments, flurbiprofen and paracetamol were dissolved in water and further diluted in media (c_stock_ = 500 µg/mL and 10 mg/mL for flurbiprofen and paracetamol, respectively).

### 4.2. Determination of mRNA Expression

The mRNA expression of ACE2 was determined as previously described [33]. Briefly, Caco-2 cells were incubated with increasing concentrations of ibuprofen (0, 1, 5, 10, 50, 100 µg/mL corresponds to 0, 4.4, 21.9, 43.8, 219, 438.1 µM), flurbiprofen (0, 0.5, 2.5, 5, 25, 50 µg/mL corresponds to 0, 2, 10.2, 20.5, 102.3, 204.7 µM), paracetamol (0, 0.25, 1.25, 2.5, 12.5, 25 µg/mL corresponds to 0, 1.7, 8.3, 16.5, 82.7, 265.4 µM), etoricoxib (0, 0.2, 1, 2, 10, 20 µg/mL corresponds to 0, 0.6, 2.8, 5.6, 27.9, 55.7 µM) or vehicle (DMSO or water) for 24 h or 48 h.

Animal tissue (Heart, Lung, Aorta) was stored in RNALater solution (Thermo Scientific, Schwerte, Germany). For mRNA isolation, dried tissue pieces were homogenized in RLT (RNA lysis buffer for cells and tissue) buffer from RNeasy Mini Kit (Qiagen, Hilden, Germany) using FastPrep-24 5G instrument (MP Biomedicals, Eschwege, Germany). Homogenates were cleared either by centrifugation (2 min, 8000× *g*) for lung and aorta tissue and by additionally using QIAshredder (Qiagen, Hilden, Germany) for heart tissue.

mRNA was extracted by RNeasy Mini Kit (Qiagen, Hilden, Germany), cDNA synthesis was performed using a First Strand cDNA-Synthesis kit (Thermo Scientific, Schwerte, Germany) including random hexamers and expression levels were determined using SYBR Select Mix (Applied Biosystems, Darmstadt, Germany) with a QuantStudio™ 12K Real-Time PCR System (Applied Biosystems, Darmstadt, Germany).

The primer for human ACE2 mRNA and β-actin detection are shown in Table 1. Relative mRNA expression was determined using the comparative CT (cycle threshold) method, normalizing relative values to the expression level of human β-actin.

For murine ACE2, β-actin, GAPDH and PPIA mRNA detection, the used primers are depicted in Table 1. Relative mRNA expression was determined using the comparative CT (cycle threshold) method, normalizing relative values to the expression level of murine β-actin, GAPDH and PPIA.

### 4.3. Determination of Protein Expression

The ACE2 protein expression was determined as previously described [33]. Briefly, Caco-2 cells were incubated with increasing concentrations of ibuprofen (0–100 µg/mL corresponds to 0–438.1 µM), flurbiprofen (0–50 µg/mL corresponds to 0–204.7 µM), paracetamol (0–25 µg/mL corresponds to 0–165.4 µm), etoricoxib (0–20 µg/mL corresponds to 0–55.7 µM) or vehicle (DMSO or water) for 24 h or 48 h. Cells were harvested and lysed in RIPA-buffer (25 mM Tris-HCl (pH7.6), 1% Sodium deoxycholate, 0.1% sodium dodecyl sulfate (SDS), 1% octylphenoxy poly(ethyleneoxy)ethanol (IPEGAL) (MP Biomedicals, Eschwege, Germany), 150 mM NaCl, Roche cOmplete™ Mini tablets (Sigma Aldrich, Schnelldorf, Germany)). Animal tissue pieces (heart, lung, aorta) were also lysed in RIPA buffer and additionally homogenized using ceramic beads, Lysing Matrix D (MP Biomedicals, Eschwege, Germany) and FastPrep-24 5G instrument (MP Biomedicals, Eschwege, Germany). The bicinchoninic acid assay (Thermo Fisher Scientific, Schwerte, Germany) was used to assess protein concentrations.

A total of 50 µg (CaCo-2 cells) or 100 µg (animal tissue) of total protein extract were separated electrophoretically by 10% SDS-PAGE and electroblotted onto nitrocellulose membranes (Amersham Life Science, Freiburg, Germany). Membranes were either blocked in 5% non-fat dry milk in TBS (Tris-buffered saline) (Carl Roth GmbH, Karlsruhe, Germany) supplemented with 0.05% Tween 20 (VWR, Darmstadt, Germany) for CaCo-2 cells or EveryBlot Blocking Buffer (Bio-Rad Laboratories, Feldkirchen, Germany) for animal tissues. All antibodies were diluted in 1% bovine serum albumin (BSA) (MP Biomedicals, Eschwege, Germany) in 0.1% Tween 20 in TBS. Membranes were incubated with the respective ACE2 (1:1000), β-Actin (1:10,000) and GAPDH (1:5000) primary antibodies overnight at 4 °C (CaCo-2 cells: both ACE2 and ß-Actin; animal tissue: ACE2) and 2 h at room temperature (animal tissue: GAPDH), washed three times with 0.05% Tween 20 in TBS, incubated with an anti-rabbit AF488 or an anti-mouse AF546 antibody (1:10,000 each) in 1% BSA in 0.1% Tween 20 in TBS for 60 min, washed again three times with 0.05% Tween 20 in TBS. The fluorescence signals were analysed on ChemiDoc™ MP Imaging System from Bio-Rad Laboratories (Feldkirchen, Germany). The mouse monoclonal anti-β-actin antibody was purchased from Sigma Aldrich (Schnelldorf, Germany), the rabbit polyclonal anti-ACE2 antibody and the mouse monoclonal anti-GAPDH from Abcam (Berlin, Germany) and the anti-rabbit AF488 and anti-mouse AF546 antibodies from Thermo Scientific (Schwerte, Germany).

### 4.4. ACE2 Activity Assay of Soluble and Membrane Bound ACE2

Supernatant from confluent Caco-2 cell culture containing serum free conditioned minimal essential medium (MEM) Eagle (phenol red) was collected. This supernatant contains soluble ACE2. Test compounds or positive control (MLN-4760) were prepared in water with a 10× higher concentration relative to the final concentration. In a black 96 well plate, 10 µL test compounds or positive control were incubated with 75 µL ACE2 substrate/assay buffer (15 µM ACE2 substrate (Mca-Ala-Pro-Lys(Dnp)-OH), 1 mM N-ethylmaleimide (NEM) (Sigma Aldrich, Schnelldorf, Germany), 1 mM phenylmethylsulfonyl fluoride (PMSF) (Carl Roth GmbH, Karlsruhe, Germany), 50 mM 2-morpholinoethanesulfonic acid sodium salt (MES Na) (Merck Chemicals GmbH, Darmstadt, Germany), 300 mM NaCl, 10 µM ZnCl_2_) and with 15 µL ACE2 containing supernatant at room temperature overnight (18 h) on a shaker (120 rpm). For the membrane-bound activity assay adapted from Xiao et al. [34], in a 96 well cell culture plate 10,000 CaCo-2 cells in phenol red MEM Eagle containing 10% FCS were incubate for 20 h. Serum containing medium was removed and 90 µL of 12.5 µM ACE2 substrate diluted in colourless RPMI medium were added. The 10 µL test compounds or positive control (prepared with a 10× higher concentration in RPMI medium) were added and incubated 18h in a 37 °C incubator with 5% CO_2_. For the soluble ACE2 activity assay, any liquid sticking to the plate seal was removed by centrifugation (4 min, 400× *g*, RT) and measurements were performed without plate seal. For the membrane-bound activity assay, the plate was centrifuged and 90 µL of supernatant were transferred to a black 96 well plate for measurement. The fluorescence with excitation wavelength of 320 nm and emission wavelength of 405 nm was measured with the Tecan Plate reader. To calculate the relative ACE2 activity, the fluorescence intensity (FI) values of the samples were related to untreated samples. The untreated samples were defined as 100% activity. The ACE2 substrate Mca-Ala-Pro-Lys(Dnp)-OH was obtained from VWR (Darmstadt, Germany).

### 4.5. Animal Study

#### 4.5.1. Test Animals

Male C57Bl/6J mice (age of ~12 weeks at the start of the study) were purchased from Charles River (Sulzfeld, Germany): 32 total, 8 mice per group. This mouse strain was selected because there are a relatively large number of studies on ibuprofen in C57Bl/6J mice and therefore it is easier to choose relevant ibuprofen doses [35,36]. The choice of gender was based on the suggestion that men appear to be more affected by COVID-19 than women [37,38,39]. All procedures were in accordance with the Principles of Laboratory Animal Care (NIH publication no. 86-23, revised 1985) and regulations of Gesellschaft für Versuchstierkunde/Society of Laboratory Animal Science (GV-SOLAS) and approved by the local Ethics Committee for Animal Research in Darmstadt (approval number F152/2000). All efforts were made to minimize animal suffering, to reduce the number of animals used, and to use alternatives to in vivo techniques, if available. Also, the vivo study was conducted according to the Enhancing Quality in Preclinical Data (EQIPD) and the ARRIVE-Guidelines. EQIPD supports robustness and reliability of preclinical biomedical research (http://www.eqipd.online/, 17 January 2022) [40]. The ITMP in vivo group has implemented this quality system in the lab and has received the EQIPD certification.

#### 4.5.2. Housing Conditions

Upon arrival, the animals were housed in GM500 Tecniplast cages for mice and a Digital ready Green line Mouse rack (DGM™) with a smartflow-ivc-air-handling-unit (Tecniplast Deutschland GmbH, Hohenpeißenberg, Germany). Animals were housed on sawdust bedding (1 animal per cage, randomization of animals occurred). The room was illuminated by lights timed to give a 12 h light dark cycle (on 07.00, off 19.00), the temperature range was 20 °C to 22 °C and the relative humidity range was 45% to 65%. The mice were allowed to acclimatize for at least 7 days before the start of the experiment and had free access to food and tap water.

#### 4.5.3. Treatment Schedule

Ibuprofen (I1892, Ibuprofen sodium salt, analytical standard, ≥98% GC (Sigma-Aldrich Chemie GmbH, Taufkirchen, Germany) was administered orally in the drinking water for a period of 7 days. Four different dose groups were compared (each group *n* = 8): 0 mg/kg, 50 mg/kg, 100 mg/kg and 200 mg/kg. The bottles were prepared fresh each day. Ibuprofen dose selection was based on clinically recommended doses [18,19]. These were approximate doses, since ibuprofen was administered in the drinking water. In order to prepare and estimate the doses as accurate as possible, the animals were weighed and the amount the mice drank was determined (by weighing the drinking bottles) each day (see Table 2). Since each animal was housed in a separate cage, the amount they drank could be measured relatively accurate.

#### 4.5.4. Sacrifice of the Animals and Blood Plasma and Organ Collection

After 7 days, following cardiac blood sampling under deep anaesthesia (following ketamine/xylazine, 180/10 mg/kg i.p., 10 mL/kg), an intracardiac perfusion with PBS solution was performed to wash the vascular system. Blood was collected in blood collection tubes containing the anticoagulant K3EDTA (haematology/potassium EDTA preparation in a round bottom inner tube, Microvette^®^, 500 μL, SARS20.1341, VWR, Darmstadt, Germany). Plasma was collected following centrifugation for 15 min at 1000× *g*, 2–8 °C within 30 min of collection. Plasma was stored in 100–200 µL aliquots or at least in 2 equal aliquots at −80 °C. For further mRNA analyses (using quantitative PCR), small pieces from each organ (heart, aorta (left ventricle or apex) and lungs) were prepared using a scalpel or scissors, and stored in 500 µL RNALater (Thermo Scientific, Schwerte, Germany). After incubation in RNALater overnight in a fridge (2–8 °C), the tissues were stored at –80°C. For further protein analyses (using Western Blot), the remaining parts of each organ were shock frozen in liquid N_2_ and then stored at −80 °C.

### 4.6. Determination of Angiotensin 1–7 in Plasma

Plasma angiotensin 1–7 levels were determined by using mouse angiotensin 1–7 (Ang1–7) ELISA Kit (Hycultec, Beutelsbach, Germany). Undiluted plasma samples were processed according to the manufacturer manual. Angiotensin 1–7 plasma concentrations were calculated using the equation from the measured standard curve.

### 4.7. Infection Assay by Spike Protein Immunostaining

The infection assay was achieved as previously described with some slight changes [33]. Briefly, confluent Caco-2 cells were preincubated with the respective compound dilutions for 1 h and subsequently infected with SARS-CoV-2 at an MOI of 0.01 for 24 h in presence of the compounds. Afterwards, the cells were, fixed, blocked and incubated with an anti-SARS-CoV-2 spike antibody (rabbit, 1:1500, SinoBiological (Eschborn, Germany)) for 1 h at 37 °C, washed twice and subsequently incubated with an HRP-coupled anti-rabbit antibody (goat, 1:1000, Jackson Immunoresearch (Cambridgeshire, UK)) for 1 h at 37 °C. After another two washing steps, the cells were stained by addition of 3-Amino-9-ethylcarbazole (AEC) solution for 10 minutes, washed and the percentage of spike positive area was detected in the BIOREADER-7000 F-z device. The percentage of spike positive area per well was quantified and the values of the compound treated samples were normalized to the virus control without compounds (=100%). Values lower than 100% represent virus inhibition, while values above 100% could indicate virus promotion.

### 4.8. Statistics

The in vitro results are presented as means ± standard errors (SEM). The data was analysed with two-way ANOVA and with Dunnett’s multiple comparisons test. For all calculations and creation of graphs, GraphPad Prism 8 (STATCON GmbH, Witzenhausen, Germany) was used and *p* < 0.05 was considered the threshold for significance. 

For the analyses of the ex vivo results, an outlier analysis was conducted to detect extreme outliers (more than three-fold the interquartile range, IQR). Outlier detection was based on SPSS (Crayon Deutschland GmbH, Unterhaching, Germany) Boxplots with extreme outliers (3 × IQR). Following outlier exclusion, it was determined if data was normally distributed and dependent on that, either a parametric (ANOVA) or non-parametric test (Kruskal-Wallis H, Mann-Whitney U) was performed (a p-value lower than 0.05 was considered statistically significant). Graphs were made using GraphPad Prism 8. The ex vivo results are presented in scatter-dot-plots with individual data points (and median ± IQR).

## Figures and Tables

**Figure 1 ijms-23-01049-f001:**
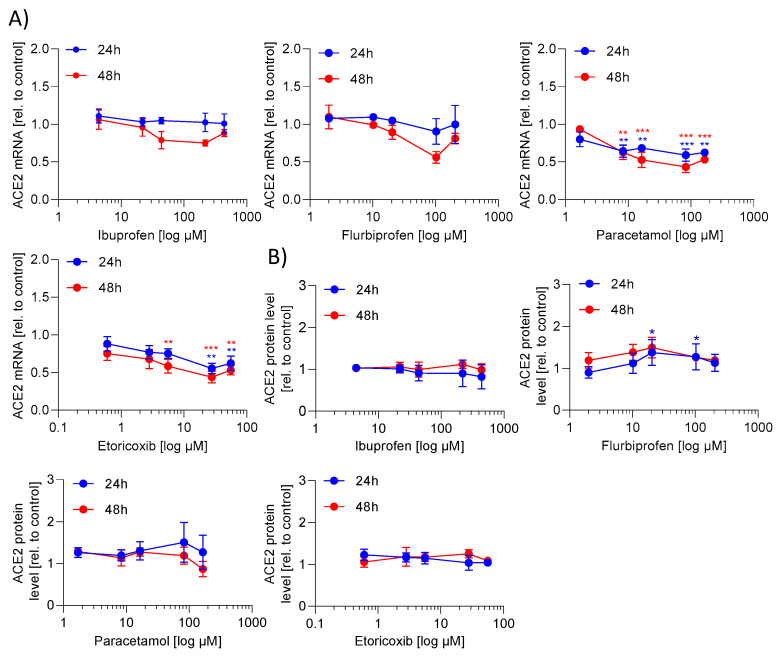
Effect of NSAIDs and paracetamol on angiotensin converting enzyme 2 (ACE2) mRNA and protein expression in Caco-2 cells. Caco-2 cells were incubated with test substances or vehicle in the indicated concentrations for 24 h or 48 h. (**A**) The mRNA expression was determined by quantitative polymerase chain reaction (PCR) and normalized to β-actin. (**B**) Protein expression determined by western blot technology was analysed by the optical densitometric analysis achieved with the Image Lab software version 6.0 (Bio-Rad Laboratories, Hercules, CA, USA). The protein expression was normalized to β-actin. To obtain the fold induction the mRNA or protein expression of drug treated samples were related to the vehicle treated samples (control). The experiment was achieved in three biological and three technical replicates. The mean of three technical replicates are shown and used for statistical analysis (two-way ANOVA with Dunnett’s multiple comparisons test). * *p* < 0.05, ** *p* < 0.01 and *** *p* < 0.001 show statistically significant difference between drug treated and vehicle treated samples.

**Figure 2 ijms-23-01049-f002:**
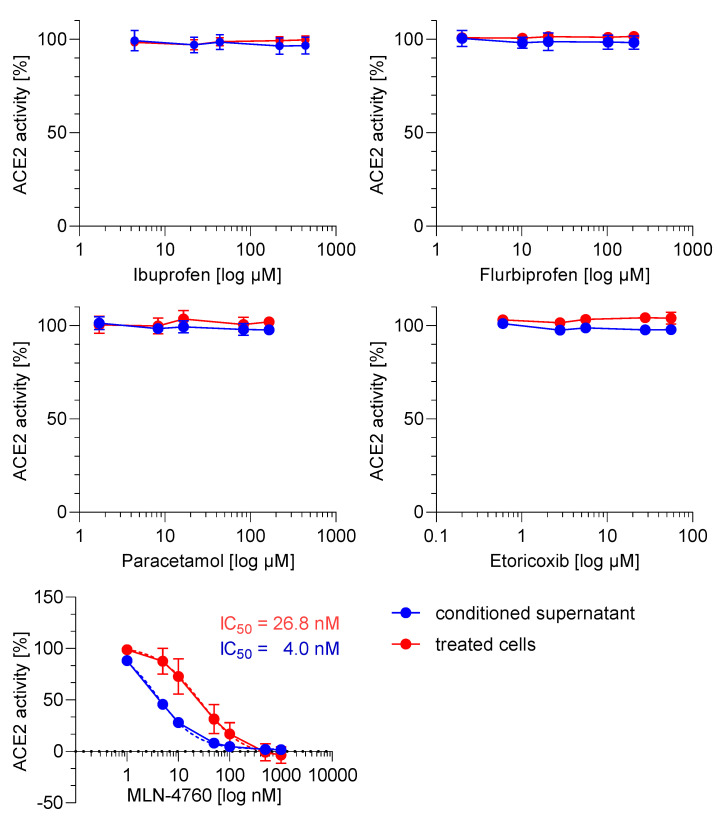
Effect of NSAIDs and paracetamol on soluble and membrane-bound angiotensin converting enzyme 2 (ACE2) activity in Caco-2 cells. For the soluble ACE activity Caco-2 supernatant and for the membrane-bound ACE2 activity Caco-2 cells were incubated with test substances or positive control (MLN-4760) for 18 h and the activity was determined with a fluorogenic substrate. To calculate the relative ACE2 activity, the fluorescence intensity (FI) values of the samples were related to untreated samples. The untreated samples were defined as 100% activity. Two-way ANOVA with multiple comparisons test was used to analyse statistical difference between drug treated and vehicle treated samples.

**Figure 3 ijms-23-01049-f003:**
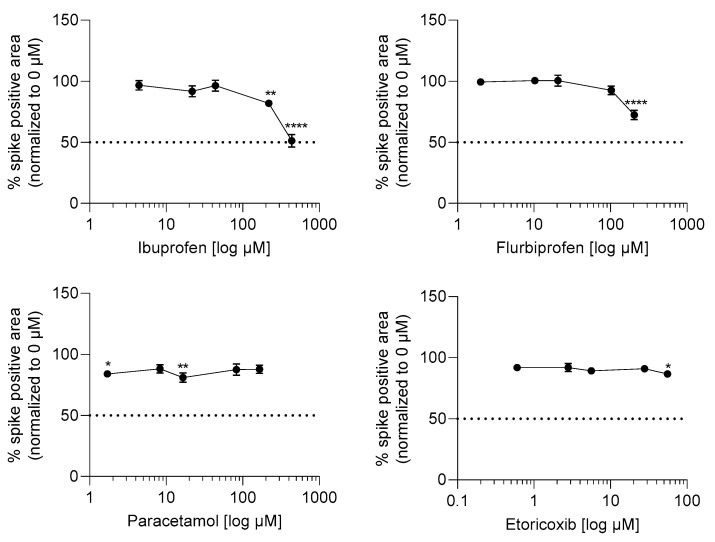
Effect of NSAIDs and paracetamol on SARS-CoV-2 virus replication. Caco-2 cells were preincubated with test substances or vehicle in the indicated concentrations for 1 h and subsequently infected with SARS-CoV-2 at an MOI of 0.01 for 24 h in the presence of compound dilutions. Afterwards, the cells were fixed and stained via immunohistochemistry against SARS-CoV-2 spike protein. The percentage of spike positive area per well was quantified and the values of the compound treated samples were normalized to the virus control without compounds (=100%). Values lower or higher than 100% represent virus inhibition or promotion, respectively. The experiment was achieved in three biological replicates. For statistical analysis, one-way ANOVA with multiple comparisons test was used. * *p* < 0.01, ** *p* < 0.01, **** *p* < 0.0001 show statistically significant differences between drug treated and vehicle treated samples.

**Figure 4 ijms-23-01049-f004:**
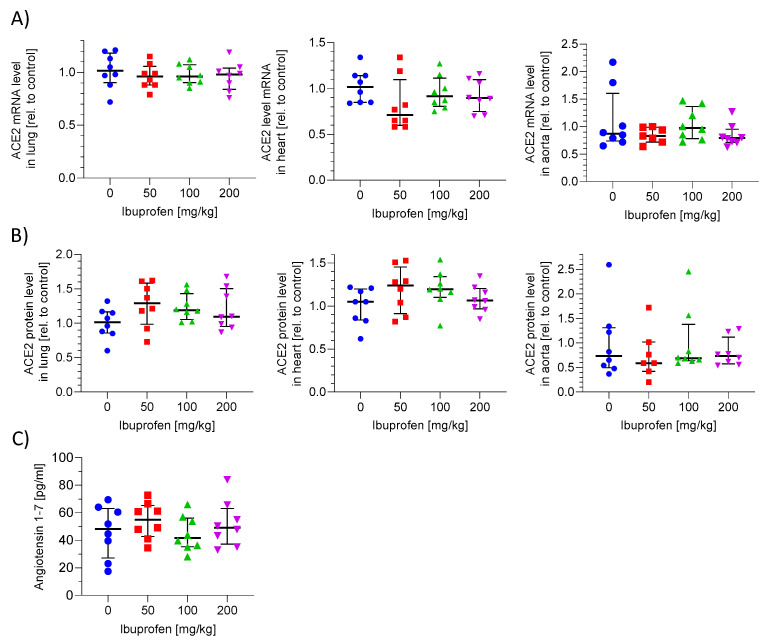
ACE2 mRNA/protein expression and activity in ibuprofen treated mice. Male C57Bl/6J mice were treated with 50, 100, 200 mg/kg ibuprofen or vehicle for 7 days. ACE2 mRNA (**A**) or protein (**B**) levels of heart, lung and aorta were determined by quantitative PCR or western blot technology, respectively. To obtain the relative values the mRNA and protein levels of treated mice were related to untreated mice. (**C**) Angiotensin 1–7 plasma levels determined with ELISA. Each treatment group consists of 8 mice.

**Table 1 ijms-23-01049-t001:** Sequences of Primer.

Primer	Sequence
human ACE-for	5′-AATGGGTCTTCAGTGCTCTC-3′
human ACE-rev	5′-GAGCCTCTCATTGTAGTC-3′
human β-actin-for	5′- CCAACCGCGAGAAGATGA-3′
human β-actin-rev	5′-CCAGAGGCGTACAGGGATAG-3′
murine ACE2-for	5′-TCCATTGGTCTTCTGCCATC-3′
murine ACE2-rev	5′-AACGATCTCCCGCTTCATCTC-3′
murine β-actin-for	5′-GGCTGTATTCCCCTCCATCG-3′
murine β-actin-rev	5′-CCAGTTGGTAACAATGCCATGT-3′
murine GAPD-for	5′-AGGTCGGTGTGAACGGATTTG-3′
murine GAPDH-rev	5′-TGTAGACCATGTAGTTGAGGTCA-3′
murine PPIA-for	5′-GAGCTGTTTGCAGACAAAGTTC-3′
murine PPIA-rev	5′-CCCTGGCACATGAATCCTGG-3′

**Table 2 ijms-23-01049-t002:** Body weight (A) and amount of daily water consumption (B), mean ± SEM, in male C57Bl/6J mice treated with 50, 100, 100 mg/kg ibuprofen or vehicle for 7 days. Each treatment group consists of 8 mice.

Dose Ibuprofen	0 mg/kg (*n* = 8)	50 mg/kg (*n* = 8)	100 mg/kg (*n* = 8)	200 mg/kg (*n* = 8)
Body weight (g) (A)				
Mean	27.9	27.3	27.4	26.8
SEM	0.14	0.23	0.15	0.33
Amount (mL) water consumption per day (B)				
Mean	5.6	5.8	5.7	5.9
SEM	0.10	0.13	0.12	0.17

## Data Availability

Data are available from the corresponding author upon reasonable request.

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
