# Peer review of "Ibuprofen, Flurbiprofen, Etoricoxib or Paracetamol Do Not Influence ACE2 Expression and Activity In Vitro or in Mice and Do Not Exacerbate In-Vitro SARS-CoV-2 Infection"

_ijms, 2022, doi:10.3390/ijms23031049_

Round 1

Reviewer 1 Report

In the manuscript entitled “Ibuprofen, other NSAIDs or acetaminophen do not influence ACE2 expression or exacerbate SARS-CoV-2 infection”, de Bruin et al. explore the proposed effect of Ibuprofen, NSAIDs or acetaminophen on ACE2 expression and find no effects but do report reduced SARS-CoV-2 replication in vitro. Overall, the manuscript is well written and the experiments robust. Although the results reported are negative in nature, the manuscript addresses an important controversy in the field that can have clinical repercussions an in this context is worthwhile to be published. Some suggestions to improve the paper follow:

When referring to drug concentrations, please restrict to only using one type of units (preferably microM) so the reader can make direct comparisons.

Regarding titrations, it is best to present them in logarithmic scales (for the concentration of the compound).

The observation regarding SARS-CoV-2 replication in vitro is interesting. Can the authors speculate regarding possible mechanisms that could be explored in future studies? Are there any known off-target effects of these drugs (outside COX) that could explain this observation?

Can the authors perform a more complete titration for Figure 3 (reaching saturation of the effect), and plot them in log scale and fit to a dose-dependent model to better validate the effect?

Author Response

Comments and Suggestions from Reviewer 1

In the manuscript entitled “Ibuprofen, other NSAIDs or acetaminophen do not influence ACE2 expression or exacerbate SARS-CoV-2 infection”, de Bruin et al. explore the proposed effect of Ibuprofen, NSAIDs or acetaminophen on ACE2 expression and find no effects but do report reduced SARS-CoV-2 replication in vitro. Overall, the manuscript is well written and the experiments robust. Although the results reported are negative in nature, the manuscript addresses an important controversy in the field that can have clinical repercussions an in this context is worthwhile to be published. Some suggestions to improve the paper follow:

When referring to drug concentrations, please restrict to only using one type of units (preferably microM) so the reader can make direct comparisons.

Response:

As suggested by the reviewer the units were changed to µM.

Regarding titrations, it is best to present them in logarithmic scales (for the concentration of the compound).

Response:

As suggested by the reviewer a logarithmic scale for compound concentration was used.

The observation regarding SARS-CoV-2 replication in vitro is interesting. Can the authors speculate regarding possible mechanisms that could be explored in future studies? Are there any known off-target effects of these drugs (outside COX) that could explain this observation?

Response:

We thank the reviewer for this comment. We have changed the Discussion section accordingly, adding the following paragraph:

As it occurs at concentrations well above the IC50 for COX, the effect of ibuprofen on viral load is likely due to off target effects of the drug. For example, PARP cleavage has been observed in colon cancer cells (HCT-15 and HCA-7) at high concentrations of ibuprofen[1]. The effect seems to be at least in part COX-2 independent, because it was observed  also in COX-2 negative HCT-15 cells. TMPRSS2 has a PARP1 binding site near to the promoter region. Since PARP1 regulates gene expression at the transcription level, Lodhi et al. speculated that virus entry can be blocked by inhibiting PARP1[2]. It is possible that PARP1 reactivates the transcription of TMPRSS2 during viral infection[3]. Thus, ibuprofen may prevent virus entry via inhibition of PARP. This speculation is supported by the finding that the PARP inhibitor stenoparib inhibits SARS-CoV-2 replication in vitro[4].

Naproxen which shows structural similarity to ibuprofen also has weak anti-viral effects that were attributed to binding to the nucleoprotein of SARS-CoV-2 [5] and it could be speculated a similar mechanism also applies for ibuprofen.

Can the authors perform a more complete titration for Figure 3 (reaching saturation of the effect), and plot them in log scale and fit to a dose-dependent model to better validate the effect?

Response:

As suggested by the reviewer, we now present the data in log scale. However, we would rather not go to higher concentrations because (a) the highest tested concentrations are already well above plasma cmax levels in patients, and (b) the solubility of the compounds is limited in water (etoricoxib even being almost insoluble in water and had to be dissolved in DMSO) and too high concentrations of DMSO are detrimental in the viral infection assay.

References:

  1. Janssen, A.; Maier, T.J.; Schiffmann, S.; Coste, O.; Seegel, M.; Geisslinger, G.; Grosch, S. Evidence of COX-2 independent induction of apoptosis and cell cycle block in human colon carcinoma cells after S- or R-ibuprofen treatment. Eur J Pharmacol 2006, 540, 24-33.
  2. Lodhi, N.; Singh, R.; Rajput, S.P.; Saquib, Q. SARS-CoV-2: Understanding the Transcriptional Regulation of ACE2 and TMPRSS2 and the Role of Single Nucleotide Polymorphism (SNP) at Codon 72 of p53 in the Innate Immune Response against Virus Infection. Int J Mol Sci 2021, 22, doi:10.3390/ijms22168660.
  3. Lodhi, N.; Kossenkov, A.V.; Tulin, A.V. Bookmarking promoters in mitotic chromatin: poly(ADP-ribose)polymerase-1 as an epigenetic mark. Nucleic acids research 2014, 42, 7028-7038, doi:10.1093/nar/gku415.
  4. Stone, N.E.; Jaramillo, S.A.; Jones, A.N.; Vazquez, A.J.; Martz, M.; Versluis, L.M.; Raniere, M.O.; Nunnally, H.E.; Zarn, K.E.; Nottingham, R.; et al. Stenoparib, an Inhibitor of Cellular Poly(ADP-Ribose) Polymerase, Blocks Replication of the SARS-CoV-2 and HCoV-NL63 Human Coronaviruses In Vitro. mBio 2021, 12, doi:10.1128/mBio.03495-20.
  5. Terrier, O.; Dilly, S.; Pizzorno, A.; Chalupska, D.; Humpolickova, J.; Boura, E.; Berenbaum, F.; Quideau, S.; Lina, B.; Feve, B.; et al. Antiviral Properties of the NSAID Drug Naproxen Targeting the Nucleoprotein of SARS-CoV-2 Coronavirus. Molecules 2021, 26, doi:10.3390/molecules26092593.

Reviewer 2 Report

The manuscript entitled: “Ibuprofen, other NSAIDs or acetaminophen do not influence ACE2 expression or exacerbate SARS-CoV-2 infection” presents the results of an interesting study aimed to evaluate in vitro and in vivo the effect of ibuprofen, flurbiprofen, etoricoxib and acetaminophen on the level of ACE2 mRNA/protein expression and activity and the influence of SARS-CoV-2 infection levels. This study is very important for the treatment of SARS-CoV-2 infection as have been many speculations related to the potential of ibuprofen and other NSAIDS drugs to facilitate viral entry and increase the disease severity.

The manuscript is well written, the material and methods are clearly described and provide enough details for the study to be reproduced.

I have some comments for improving the manuscript:

  1. Line 274 define FCS and add the manufacturer
  2. Also for all the reagents used indicate the manufacturer, eg: L-glutamine, penicillin/streptomycin
  3. Define DMSO line 280
  4. Define RLT buffer, line 293
  5. I suggest adding a table for more clarity with the primer used for ACE2 mRNA detection
  6. Define TBS, line 333
  7. Define MEM Eagle, line 348
  8. In discussion add the limitations and the strengths of the study
  9. A flowchart of the study will help the reader to understand better the study design.

ijms-1537498, Ibuprofen, other NSAIDs or acetaminophen do not influence ACE2 expression or exacerbate SARS-CoV-2 infection, presents a very interesting research focused on an important and urgent problem, finding treatment solutions for COVID19 patients. The manuscript has a good value, but some small changes are needed before publication.

The title and the overall use of acetaminophen needs to be changed with the correct INN (International Nonproprietary Name) of the drug, paracetamol

There are some other recent and valuable works that should be referenced by the authors, works that could help the readers better understand the problem of COVID19 treatment and prospects. Here are some examples that could be useful:

Comprehensive analysis of drugs to treat SARS‑CoV‑2 infection: Mechanistic insights into current COVID‑19 therapies (Review), Int J Mol Med 2020, https://doi.org/10.3892/ijmm.2020.4608

SARS-CoV-2: Repurposed Drugs and Novel Therapeutic Approaches-Insights into Chemical Structure-Biological Activity and Toxicological Screening, J Clin Med 2020 Jul 2;9(7):E2084. doi: 10.3390/jcm9072084.

The discussion should be more objective. The authors should not generalize. They tested only 3 NSAIDS: ibuprofen, flurbiprofen, etoricoxib. The many other NSAIDS with various structures that could interact in a different manner in rapport with ACE2 mRNA/protein expression and its activity.

On row 241, the authors hypothesize that only NSAIDs which act as both COX1 and COX2 inhibitors reduce viral load. This hypothesis is based on only 3 drugs. It could be just the fact that ibuprofen and flurbiprofen are propionic acid derivatives and etoricoxib is not. The example offered as “argument”, naproxen, is also a propionic acid derivative. This could indicate that the propionic acid derivatives can reduce the viral load. I find the authors hypothesis dangerous and I advise them to remove this section.

Author Response

Comments of Reviewer 2

The manuscript entitled: “Ibuprofen, other NSAIDs or acetaminophen do not influence ACE2 expression or exacerbate SARS-CoV-2 infection” presents the results of an interesting study aimed to evaluate in vitro and in vivo the effect of ibuprofen, flurbiprofen, etoricoxib and acetaminophen on the level of ACE2 mRNA/protein expression and activity and the influence of SARS-CoV-2 infection levels. This study is very important for the treatment of SARS-CoV-2 infection as have been many speculations related to the potential of ibuprofen and other NSAIDS drugs to facilitate viral entry and increase the disease severity.

The manuscript is well written, the material and methods are clearly described and provide enough details for the study to be reproduced.

I have some comments for improving the manuscript:

1.Line 274 define FCS and add the manufacturer

2.Also for all the reagents used indicate the manufacturer, eg: L-glutamine, penicillin/streptomycin

3.Define DMSO line 280

4.Define RLT buffer, line 293

5.I suggest adding a table for more clarity with the primer used for ACE2 mRNA detection

6.Define TBS, line 333

7.Define MEM Eagle, line 348

8.In discussion add the limitations and the strengths of the study

9.A flowchart of the study will help the reader to understand better the study design

Response:

As the reviewer suggested, we defined the abbreviations, added the supplier, a primer table, a flow chart of the study design and provided the limitations and strengths of the study in the discussion part.

The strength of our study is that we compared the effect of ibuprofen on ACE2 expression/activity in vitro and in mice, which allows us to conclude that ibuprofen does not facilitate virus entry via an upregulation of ACE2. A further strength is that we compared across a broad range of drug concentrations which allowed us to detect the weak in vitro antiviral activity at higher concentrations. However, the limitation of our study is the lack of clinical data. Therefore, whether this has any significance for patients with COVID 19, can only be established through clinical trials.

ijms-1537498, Ibuprofen, other NSAIDs or acetaminophen do not influence ACE2 expression or exacerbate SARS-CoV-2 infection, presents a very interesting research focused on an important and urgent problem, finding treatment solutions for COVID19 patients. The manuscript has a good value, but some small changes are needed before publication.

The title and the overall use of acetaminophen needs to be changed with the correct INN (International Nonproprietary Name) of the drug, paracetamol

Response:

As the reviewer suggested we changed the name of acetaminophen to paracetamol.

There are some other recent and valuable works that should be referenced by the authors, works that could help the readers better understand the problem of COVID19 treatment and prospects. Here are some examples that could be useful:

Comprehensive analysis of drugs to treat SARS‑CoV‑2 infection: Mechanistic insights into current COVID‑19 therapies (Review), Int J Mol Med 2020, https://doi.org/10.3892/ijmm.2020.4608

SARS-CoV-2: Repurposed Drugs and Novel Therapeutic Approaches-Insights into Chemical Structure-Biological Activity and Toxicological Screening, J Clin Med 2020 Jul 2;9(7):E2084. doi: 10.3390/jcm9072084.

Response:

As the reviewer suggested we added in the discussion potential treatment strategies for COVID19 (see page 8).

The discussion should be more objective. The authors should not generalize. They tested only 3 NSAIDS: ibuprofen, flurbiprofen, etoricoxib. The many other NSAIDS with various structures that could interact in a different manner in rapport with ACE2 mRNA/protein expression and its activity.

Response:

As the reviewer suggested we avoided to sum up flurbiprofen, ibuprofen and etoricoxib as NSAIDs in the discussion. Moreover, we stated in the conclusion “However, we cannot exclude the possibility that other NSAIDs may affect ACE2 expression and activity” (see page 8).

On row 241, the authors hypothesize that only NSAIDs which act as both COX1 and COX2 inhibitors reduce viral load. This hypothesis is based on only 3 drugs. It could be just the fact that ibuprofen and flurbiprofen are propionic acid derivatives and etoricoxib is not. The example offered as “argument”, naproxen, is also a propionic acid derivative. This could indicate that the propionic acid derivatives can reduce the viral load. I find the authors hypothesis dangerous and I advise them to remove this section.

Response:

We agree with the reviewer and removed this hypothesis.

Reviewer 3 Report

The title must be changed so as not to give the impression that the results are directly applicable to humans. Since the mice were not infected, technically the title is very remote extrapolation of the findings. 

I note that the authors declared that some of them are employees of Reckitt, the owners and distributors of the Nurofen. They should be especially careful when concluding from the in vitro and animal experiments so as to avoid perceived bias. 

Author Response

Comments and Suggestions from Reviewer 3

The title must be changed so as not to give the impression that the results are directly applicable to humans. Since the mice were not infected, technically the title is very remote extrapolation of the findings.

Response:

As the reviewer suggested we changed the title to “Ibuprofen, flurbiprofen, etoricoxib or paracetamol do not influence ACE2 expression and activity in vitro or in mice and do not exacerbate in-vitro SARS-CoV-2 infection”.

I note that the authors declared that some of them are employees of Reckitt, the owners and distributors of the Nurofen. They should be especially careful when concluding from the in vitro and animal experiments so as to avoid perceived bias.

Response:

We are aware of this potential conflicting interest and were very careful in (a) making it transparent in the Authors’ note and (b) avoiding biased interpretations or conclusions of our experimental data. As an example, we stated clearly at the end of our conclusion “Whether this has any significance for patients with COVID 19, can only be established through clinical trials”.

Round 2

Reviewer 1 Report

I thank the authors for addressing my concerns. I believe that the manuscript has improved.

Reviewer 2 Report

The author addressed all my comments. The manuscript is much improved and ready for acceptance. 

Reviewer 3 Report

Thank you for the revised version. The revised title is more accurate than previous. I hope this will be an important addition to the growing literature.